

# Prevalence of abnormal urinary cadmium and risk of albuminuria as a primary bioindicator for kidney problems among a healthy population

Mohd Faizal Madrim[1,2], Mohd Hasni Ja'afar[1] and Rozita Hod[1]

[1] Department of Community Health, Faculty of Medicine, Universiti Kebangsaan Malaysia, Kuala Lumpur, Federal Territory of Kuala Lumpur, Malaysia

[2] Department of Public Health Medicine, Faculty of Medicine and Health Sciences, Universiti Malaysia Sabah, Kota Kinabalu, Sabah, Malaysia

## ABSTRACT

**Background**. The prevalence of chronic kidney disease is increasing globally, ranking 27th as the cause of death in the 1990s, rising to 18th in 2010 and 10th in 2019. Non-communicable diseases such as diabetes and hypertension have been identified as the common contributing factors, while there is also evidence linking environmental pollutants, especially cadmium, to kidney disease. This study aimed at investigating the level of urinary cadmium and its relationship to albuminuria as an early indicator of kidney problems in the Kepong community.

**Methods**. Respondents were surveyed as part of several health-related programs organized by the Kepong District Health Office involving local communities in and around the district from April 2019 to December 2019. Urinalysis of two urine samples was carried out using a Mission reagent strip and an Inductively Coupled Plasma Mass Spectrometry (ICP-MS) test to detect the presence and level of urinary cadmium.

**Results**. A total of 240 respondents were enrolled from April 2019 to December 2019. Urinalysis of two urine samples was carried out using a Mission reagent strip and an Inductively Coupled Plasma Mass Spectrometry (ICP-MS) test to detect the level of urinary cadmium. The respondents' average age was 41-year-old ($\pm 13.23$). Among them, 49.6% were male, 85.0% Malay, 5.8% Chinese and 8.3% Indian. 55.0% had background of tertiary, 39.6% secondary and 5.4% primary level of education. 52.1% were categorized in B40, 34.6% in M40 and 13.3% in T20 based on monthly household income category. 26.7% were hypertensive, 6.7% diabetic, 4.2% had dyslipidemia, 51.7% had urinary cadmium above the alert level, and 27.1% had albuminuria.

**Discussion**. Risk factors for albuminuria that have been identified are age with adjusted odds ratio (AOR) 3.53 (1.41–8.83; $p < 0.05$), highest educational level with AOR 2.18 (1.14–4.17; $p < 0.05$), diabetes with AOR 3.36 (1.07–10.52; $p < 0.05$), and urinary cadmium with AOR 4.72 (2.33–9.59; $p < 0.001$), with future screening programs placing greater attention to those at risk and further research is required to determine the cause of exposure to cadmium.

Corresponding author
Mohd Hasni Ja'afar,
drmhasni@ukm.edu.my,
drmhasni1965@gmail.com

## INTRODUCTION

Chronic kidney disease (CKD) is a major non-communicable disease that contributes significantly to morbidity and mortality globally. One of the aims outlined in the UN's Sustainable Development Goal (SDG) is to decrease premature deaths from non-communicable diseases by one-third by 2030 and in order to achieve it, greater attention has to be given to this disease.

The number of people seeking treatment for this disease is increasing, with more than 2.5 million current reported. According to projections, the number could reach 5.4 million in a decade (*Liyanage et al., 2015*). Malaysia is experiencing a similar magnitude of the problem, where there was an 11-fold increase in the number of patients receiving hemodialysis treatment registered in the 22nd Malaysian Dialysis and Transplant Registry over a 10-year period (2004–2014) (*Goh, Ong & Lim, 2015*).

In general, there are two groups of risk factors contributing to the development of this disease: traditional and non-traditional. The traditional group mainly consists of metabolic disorders such as hypertension, diabetes mellitus, and obesity, while the non-traditional group consists of infections and drug-related disorders. There have been previous reports of CKD cases that were not related to either traditional or non-traditional risk factors (*Abumwais, 2012*; *Athuraliya et al., 2011*) but instead linked to environmental toxicants, specifically cadmium (Cd). Cd has specific properties that enable it to accumulate in kidneys and disrupt the system. Following exposure, Cd create a complex with metallothionein which initially filtered by glomerulus and subsequently reabsorbed at proximal and distal tubules. Upon entering tubular cells, lysosome breaks the complex to free Cd and initiates damage to kidneys. The process here includes oxidative stress, inflammatory cell infiltration and downregulating mitochondrial coenzymes (*Rana, Tangpong & Rahman, 2018*; *Naji et al., 2014*; *Saif et al., 2015*; *Johri, Jacquillet & Unwin, 2010*).

CKD is unlikely to exhibit obvious features or symptoms, especially in the early stages, increasing the likelihood for it to be overlooked and albuminuria is one of the most basic tests for detecting early signs of CKD (*Levey, Becker & Inker, 2015*). As there is still a lack of local awareness about the issue, this study aimed to determine the prevalence of high urinary Cd levels and assess their relationship with albuminuria as an early indicator of kidney disease in the Kepong, Klang Valley population.

## MATERIALS & METHODS

The general population of Kepong, Kuala Lumpur was selected for this cross-sectional study, and samples were collected from Kepong residents aged 18 years and above of both genders residing in Kepong, with informed consent obtained from every participant using a consent form for this study. The inclusion criteria were: (a) Malaysian nationality or permanent resident; (b) residing in Kepong for at least 10 years; and (c) consented for this study, whereby the exclusion critera were: (a) respondents with leucocytes $\geq$ 1+ or positive nitrite detected in the dipstick urinalysis; (b) respondents whose urine sample showed red blood cells $\geq$ 1+; respondents who had known kidney disease; and (c) respondents who failed to converse or understand Malay or English language.

Every participant completed a self-administered questionnaire that included demographic, socioeconomic, smoking status, exercise, and also medical history. The participants' blood pressure (BP), heart rate, body mass index (BMI), blood glucose, and dipstick urinalysis were all measured on-site.

Weight was measured using the 'SECA 813 Digital High-Capacity Floor Scale, Japan', height was measured using the 'SECA 217 Stadiometer, Japan', blood pressure was measured using the 'Omron Digital Automatic Blood Pressure Monitor Model HEM-907, Omron Healthcare Co Ltd, Kyoto, Japan', and capillary blood glucose was measured using the Accu-Chek Performa 2 Glucometer.

Two separate disposable polypropylene containers were used to collect clean-catch, mid-stream urine specimens. The first container was for dipstick urinalysis (Mission Urinalysis Reagent Strips,) at the study site, while the second one was for a Cd check using Inductively Coupled Plasma Mass Spectrometry (ICP-MS) at the environmental laboratory in Universiti Kebangsaan Malaysia.

## Definitions

Urinary albumin-to-creatinine ratio (ACR, milligram per gram (mg/g)) readings as the output of the dipstick urinalysis were used to define albuminuria. According to the guideline of the American Diabetes Association, microalbuminuria and macroalbuminuria were defined as increases in ACR between 30 and 299 mg/g and 300 mg/g or higher, respectively, and referred to as 'albuminuria' in this study.

High urinary Cd levels were determined using the Agency for Toxic Substances and Disease Registry (ATSDR) and European Food Safety Authority (EFSA) reference values (*U.S. Department of Health and Human Services, 2012*; *EFSA Panel on Contaminants in the Food Chain (CONTAM), 2011*), with readings greater than 0.959 $\mu$g/L considered high levels of urinary Cd.

Systolic BP $\geq$ 140 mmHg or diastolic BP $\geq$ 90 mmHg or self-reported diagnosis of high BP were used to define hypertension. Those who had a history of diabetes mellitus or a fasting capillary blood glucose $\geq$ 7.0 mmol/L or non-fasting capillary blood glucose $\geq$ 11.1 mmol/L (*Diabetes, 2013*) were described as having diabetes mellitus. Participants were deemed to have fasted if they had not eaten anything for at least 8 h before the blood test. Self-reported diagnosis of deranged blood cholesterol level was defined as dyslipidemia and a body mass index (BMI) of $\geq$ 30 kg/m (*National Cholesterol Education Program (NCEP), 2002*) was defined as obesity.

## Statistical analysis

Analysis was carried out on data obtained from adults aged 18 years and older participating in the screening program from April 2019 to October 2019, with those having incomplete data excluded. Analysis was performed using Statistical Package for Social Sciences (SPSS) version 25. The normally distributed continuous variables were summarized using the mean and standard deviation (SD), while the non-normally distributed variables were expressed using the median and interquartile range (IQR). For categorical variables, the frequencies and percentages (%) were tabulated. Univariate logistic regression was used to

determine the unadjusted odds ratio (OR) for albuminuria and the simultaneous effects of various risk factors, when adjusted for other confounders, were determined using multivariable logistic regression.

This National University of Malaysia (UKM) granted Ethical approval to carry out the study. (Ethical Approval Ref: UKM FPR.4/244/FF-2019-100, 22nd March 2019).

## RESULTS

A total of 240 people participated in this study. Table 1 displays the sample population's characteristics and the participants were comprised of 50.4% females, 85.0% ethnic Malays, 5.8% Chinese, 8.3% Indians, and 0.8% others, with an average age of 41.4 years. Among them, 45.0% did not have a high education level background and 52.1% were in the B40 group (bottom 40% of households with monthly income of RM3,152 and below according to Department of Statistics Malaysia 2019). Hypertension was present in 14.6% of the participants, diabetes in 6.7% of the participants, dyslipidemia in 4.2% of the participants, and obesity in 44.2% of the participants, with no fasting respondents when their blood was taken for sugar check.

The ICP-MS 240 was used to screen Cd concentrations of urine samples and the detection limit for Graphite Furnace Atomic Absorption Spectrometry (GFAAS) was as low as 0.1 $\mu$g / L urine and the sensitivity of this analysis is 99%.. High urinary Cd levels were present in 51.7% of the participants and albuminuria was detected in 27.1% of the participants.

### Bivariate analysis

Table 2 shows that the geometric mean of urinary Cd is higher in females, Indians, and unemployed people as compared to others. However, only the gender was found out to be statistically significant, with a $p$-value of 0.008.

As shown in Table 3, a Pearson correlation test was performed and there was a moderate positive correlation ($r = 0.386$) between age and urinary Cd level, but a weak positive correlation ($r = 0.154$) between duration of stay in Kepong and urinary Cd level.

According to Table 4, there were no significant associations between albuminuria and gender, ethnicity, working status, smoking status, obese and dyslipidemia. The proportion of those aged 60 years and above, with a low education background, a low household income, were hypertensive, diabetic, and had high urinary Cd with albuminuria was higher as compared to others.

### Multivariate analysis

Table 5 shows the adjusted odds ratio for albuminuria, with the adjusted analysis retaining all the variables. Individuals with high urinary Cd levels displayed the highest risk for proteinuria, followed by those aged 60 years old and above, who had diabetes, and without a high education background.

**Table 1  Sociodemographic and health profiles of participants.**

| Variables | Frequency (%) | Min ± SD | Median (Interquartile range) | Range |
|---|---|---|---|---|
| **Gender** | | | | |
| Male | 119 (49.6) | | | |
| Female | 121 (50.4) | | | |
| **Age (years)** | | 41.41 ± 13.23 | | 18–74 |
| <60 | 213 (88.8) | | | |
| ≥60 | 27 (11.2) | | | |
| **Ethnicity** | | | | |
| Malay | 204 (85.0) | | | |
| Chinese | 14 (5.8) | | | |
| Indian | 20 (8.3) | | | |
| Others | 2 (0.8) | | | |
| **Education level** | | | | |
| Primary | 13 (5.4) | | | |
| Secondary | 95 (39.6) | | | |
| Tertiary | 132 (55.0) | | | |
| **Working status** | | | | |
| Working | 156 (65.0) | | | |
| Not working | 84 (35.0) | | | |
| **Monthly household income category** | | | 3000.00 (1500.00) | 600.00–20000.00 |
| B40 | 125 (52.1) | | | |
| M40 | 83 (34.6) | | | |
| T20 | 32 (13.3) | | | |
| **Duration of staying in Kepong (years)** | | 23.82 ± 12.21 | | 5–63 |
| **Smoking status** | | | | |
| Smoking | 104 (43.3) | | | |
| Not smoking | 136 (56.7) | | | |
| **Known case of hypertension** | | | | |
| Yes | 19 (7.9) | | | |
| No | 221 (92.1) | | | |
| **Known case of diabetes** | | | | |
| Yes | 11 (4.6) | | | |
| No | 229 (95.4) | | | |
| **Known case of dyslipidemia** | | | | |
| Yes | 10 (4.2) | | | |
| No | 230 (95.8) | | | |
| **Body mass index (kg/m2)** | | 26.85 ± 5.91 | | 15.56 –53.35 |
| **Obese** | | | | |
| Yes | 106 (44.2) | | | |
| No | 134 (55.8) | | | |

 

**Table 1** (*continued*)

| Variables | Frequency (%) | Min ± SD | Median (Interquartile range) | Range |
|---|---|---|---|---|
| **Blood pressure (mmHg)** | | | | |
| **Systolic** | | 123.82 ± 14.55 | | 86–168 |
| **Systolic ≥ 140 mmHg (Yes)** | 35 (14.6) | | | |
| **Diastolic** | | 79.10 ± 9.44 | | 54–110 |
| **Diastolic ≥ 90 mmHg (Yes)** | 34 (14.2) | | | |
| **Blood sugar level (mmol/L)** | | | 6.0 (5.3) | 3.9–21.1 |

**Table 2** Urinary cadmium level of participants.

| Variables | Frequency | Geometric mean (µg/L) (SD) | Statistical tests |
|---|---|---|---|
| **Gender** | | | $t = 2.654$ |
| Male | 119 | 0.809 (1.910) | $p = 0.008$ |
| Female | 121 | 1.062 (2.399) | |
| **Ethnicity** | | | |
| Malay | 204 | 0.914 (2.188) | $F = 0.490$ |
| Chines | 14 | 0.946 (1.535) | $p = 0.689$ |
| Indian | 20 | 1.097 (3.155) | |
| Others | 2 | 0.615 (1.035) | |
| **Working status** | | | |
| Working | 156 | 0.911 (2.170) | $t = 0.472$ |
| Not working | 84 | 0.959 (2.334) | $p = 0.637$ |

**Table 3** Correlation between urinary cadmium and continuous variables.

| Variables | Urine cadmium level (µg/L) | |
|---|---|---|
| | Correlation coefficient, r | Nilai p |
| Age (years) | 0.386[**] | <0.001 |
| Duration of staying in Kepong (years) | 0.154[*] | 0.017 |

Notes.
[**]*p*-value of < 0.01 is considered significant
[*]*p*-value of < 0.05 is considered significant

## DISCUSSION

Based on the guidelines provided by ATSDR and EFSA, 51.7% of the recruited samples for this study had high urinary Cd levels, which was different from the findings of previous studies that have also been conducted in Malaysia. *Adnan et al. (2012)* reported a 14.7% prevalence of high urinary Cd levels among adults residing in Tanjung Karang, Selangor, and that the variations in prevalence were influenced by reference source differences. The ATSDR and EFSA guidelines, which focused more on kidney damage, were used in this study, while *Adnan et al. (2012)* used the Michigan Occupational Safety and Health Administration (MIOSHA) as a reference, with 2 µg/L as the action threshold level, which
**Table 4  Simple logistic regression.**

| Variables | Frequency | Albuminuria | | X2 (df) | P-value |
|---|---|---|---|---|---|
| | | **Yes** | **No** | | |
| **Gender** | | | | | |
| Male | 119 | 31 (26.1%) | 88 (73.9%) | 0.13 (1) | 0.721 |
| Female | 121 | 34 (28.1%) | 87 (71.9%) | | |
| **Ethnicity** | | | | | |
| Malay | 204 | 55 (27.0%) | 149 (73.0%) | 3.60 (3) | 0.345a |
| Chinese | 14 | 2 (14.3%) | 12 (85.7%) | | |
| Indian | 20 | 8 (40.0%) | 12 (60.0%) | | |
| Others | 2 | 0 (0.0%) | 2 (100.0%) | | |
| **Age (years)** | | | | | |
| <60 | 213 | 49 (23.0%) | 164 (77.0%) | 15.95 (1) | <0.001 |
| ≥60 | 27 | 16 (59.3%) | 11 (40.7%) | | |
| **High education background?** | | | | | |
| Yes | 132 | 23 (17.4%) | 109 (82.6%) | 13.86 (1) | <0.001 |
| No | 108 | 42 (38.9%) | 66 (61.1%) | | |
| **Working status** | | | | | |
| Not working | 84 | 28 (33.3%) | 56 (66.7%) | 2.56 (1) | 0.110 |
| Working | 156 | 37 (23.7%) | 119 (76.3%) | | |
| **Monthly household income –B40?** | | | | | |
| No | 115 | 19 (16.5%) | 96 (83.5%) | 12.47 (1) | <0.001 |
| Yes | 125 | 46 (36.8%) | 79 (63.2%) | | |
| **Smoking status** | | | | | |
| Not smoking | 136 | 36 (26.5%) | 100 (73.5%) | 0.06 (1) | 0.807 |
| Smoking | 104 | 29 (27.9%) | 75 (72.1%) | | |
| **Obese** | | | | | |
| No | 134 | 31 (23.1%) | 103 (76.9%) | 2.40 (1) | 0.122 |
| Yes | 106 | 34 (32.1%) | 72 (67.9%) | | |
| **Hypertensive** | | | | | |
| No | 176 | 38 (21.6%) | 138 (78.4%) | 10.08 (1) | 0.001 |
| Yes | 64 | 27 (42.2%) | 37 (57.8%) | | |
| **Diabetes** | | | | | |
| No | 224 | 55 (24.6%) | 169 (75.4%) | 10.89 (1) | 0.002 |
| Yes | 16 | 10 (62.5%) | 6 (37.5%) | | |
| **Dyslipidaemia** | | | | | |
| No | 230 | 63 (27.4%) | 167 (72.6%) | 0.27 (1) | 1.000 |
| Yes | 10 | 2 (20.0%) | 8 (80.0%) | | |
| **Urine cadmium level** | | | | | |
| Low | 116 | 13 (11.2%) | 103 (88.8%) | 28.66 (1) | <0.001 |
| High | 124 | 52 (41.9%) | 72 (58.1%) | | |

**Table 5  Multiple logistic regression.**

| Variables | Adjusted odds ratio | 95% confidence interval | X2 (df) | *P*-value |
|---|---|---|---|---|
| **Age (years)** | | | | |
| <60 | 1 | | | |
| ≥60 | 3.53 | 1.41; 8.83 | 7.28 (1) | 0.007 |
| **High education** | | | | |
| Yes | 1 | | | |
| No | 2.18 | 1.14; 4.17 | 5.51 (1) | 0.019 |
| **Diabetic** | | | | |
| No | 1 | | | |
| Yes | 3.36 | 1.07; 10.52 | 4.33 (1) | 0.038 |
| **Urine cadmium level** | | | | |
| Low | 1 | | | |
| High | 4.72 | 2.33; 9.59 | 18.47 (1) | <0.001 |

was approximately twice the reference level used in this study. In addition, this difference was also most likely influenced by the study area as Kepong may have more Cd pollution sources compared to Tanjung Karang, causing the prevalence of Cd urine readings to be higher in Klang Valley.

The prevalence of albuminuria for this study was 27.1%, which fell within the range of previous studies' prevalence of albuminuria of 19.5%–33.2% (*An et al., 2014*; *Chen et al., 2014*; *Kweon et al., 2012*; *Okpere, Anochie & Eke, 2012*; *Zacharias et al., 2012*; *Zheng et al., 2013*; *Zhuo et al., 2020*). There are two main factors contributing to the different findings on the prevalence of albuminuria between this study and previous studies. First, is the selection of study respondents based on different acceptance and rejection criteria. For example, according to *Chen et al. (2014)*, one of the rejection criteria of a respondent is someone who has been diagnosed with diabetes by a doctor and people with a history of diabetes were more likely to have albuminuria than others. This explains, to some extent, why the prevalence of albuminuria in this study was slightly greater compared to the finding of *Chen et al. (2014)*. The second factor is the differences between the results on the prevalence of albuminuria were due to the albuminuria test itself. Previous studies have stated that several factors must be considered when testing for albuminuria, including the type of urine sample obtained, whether it is the initial flow, intermediate flow, or final flow, the unit of measurement used, whether it is uniform with any major references or not, and whether the albuminuria test should be repeated (*Chiang et al., 2011*; *Lu et al., 2007*). Besides the two factors mentioned above, researchers have also identified other factors that may influence prevalence differences for albuminuria, such as racial or ethnic differences (*Kenealy et al., 2012*) and diurnal fluctuations in albumin excretion (*Miller et al., 2009*).

It was discovered in this study, after controlling for confounding with Multiple Logistic Regression analysis, that age, highest level of education, diabetes, and high urinary Cd levels were significant predictive factors of albuminuria with a *p*-value of less than 0.05. Although the toxic effects of heavy metals, including Cd, on renal function, have been

well documented (*Johri, Jacquillet & Unwin, 2010*; *Soderl et al., 2010*; *Evans & Elinder, 2011*; *Järup, 2003*), there is still a lack of information on the associations between Cd and albuminuria in the general Malaysian population. It is important to understand that age has substantial influence on nephrotoxicity (*El-Arabey, 2015*). In previous cross-sectional studies, the commonly used "biomarkers" associated with renal disease were a reduced estimated glomerular filtration rate (eGFR < 60 mL/min/1.73 m2) and albuminuria (urinary ACR ≥ 30 mg/g) (*Chung et al., 2014*; *Weaver et al., 2009*; *Buser et al., 2016*; *Grau-Perez et al., 2017*), although some authors referred to CKD as eGFR or urinary ACR (*Kim et al., 2015*). This study reveals a significant risk for diabetes and albuminuria (adjusted odds ratio (aOR) 3.36, 95% CI [1.07–10.52]). This is consistent with the findings from previous study involving Asian respondents (*Kweon et al., 2012*; *Satirapoj et al., 2011*). It is well known that the pathogenesis of diabetic nephropathy is related with renal fibrosis that can be caused by several factors namely renal haemodynamic changes, glucose metabolism abnormalities-associated oxidative stress, inflammatory processes and overactive renin-angiotensin-aldosterone system (*Melmed et al., 2015*). In addition, based on recent molecular studies, the pathogenesis is associated with genetic and epigenetic regulation, mitochondria dysfunction and podocyte autophagy (*Lin et al., 2018*).

The relationship between urinary Cd and decreased GFR or albuminuria in adult participants of the United States NHANES 2007–2012 was investigated by *Buser et al. (2016)* using multivariate analysis and it was discovered that there was an important positive correlation between albuminuria and urinary Cd. Furthermore, *Grau-Perez et al. (2017)* examined the relationship between urinary Cd levels and albuminuria in adult participants who were part of the Hortega Study, including a general Spanish population, and discovered using a multivariate analysis that increased urinary Cd levels were significantly correlated with albuminuria. Findings from both cross-sectional studies (*Buser et al., 2016*; *Grau-Perez et al., 2017*) were similar to the findings of this study, where there was a significant correlation between urinary Cd levels and albuminuria.

The results may be affected by ethnic and cultural differences between studies, such as local diets, as well as variations in study designs. The present study, which discovered that Malays had a higher risk of developing renal disease compared to others, highlighted the impact of ethnic background on the risk of developing renal disease. Cd may be a contaminant in drinking water and crops have been shown to absorb Cd from polluted soil and water; hence, variations in regional water and soil contamination can affect outcomes. Furthermore, it is imperative to consider biomarkers other than albuminuria that may be better at determining the correlations between heavy metal exposure and renal disease.

The variations in the potential effects of Cd on renal disease development identified by the studies listed above and the current study could be due to differences in study designs, study populations, and sensitivity of analyses. These disparities can reflect heterogeneity in the measures and methods that were used to detect renal disease (*e.g.*, eGFR, albuminuria, or proteinuria), which may have varying degrees of sensitivity for the detection of renal disease. A study comparing proteinuria to albuminuria as CKD biomarkers found that there was a significant correlation between 24-hour proteinuria for protein to creatinine in urine and albumin-to-creatinine ratio (ACR) and that both protein-to-creatinine ratio

(PCR) and ACR were suitable biomarkers for cardiovascular events, renal dysfunction, or mortality (*Guh, 2010*). However, studies like this one that evaluate the correlations between the biological levels of heavy metals and a single potential renal disease biomarker like albuminuria can only conclude that there is an association between specific heavy metals with a certain biomarker, not the influence of heavy metals on renal disease development.

There are several limitations to this study that should be taken into consideration during the interpretation of results. This study's cross-sectional design precludes the capability to determine causality and cross-sectional data alone cannot adequately evaluate the complex association between low-level biological Cd exposure and renal disease. Animal and clinical studies, particularly longitudinal studies, are required for further evaluation of the relationship between heavy metals exposure and renal disease development. The cross-sectional analysis could not confirm whether low-level exposure to Cd led to actual kidney damage in people with albuminuria. Furthermore, besides renal disease, other conditions may be the cause of albuminuria, which possibly confounded our findings. *Zhang et al. (2015)* recently suggested that N-acetyl-β-D- glucosaminidase, metallothionein, and alkaline phosphatase were sensitive biomarkers for evaluating long-term Cd exposure. There is a possibility that besides albuminuria, other biomarkers could prove to be more sensitive for the evaluation of low-level heavy metal exposure effects on renal function.

## CONCLUSIONS

In conclusion, it has been shown through the multivariate results of this study that there was a significant association between urinary Cd levels and albuminuria in Kepong, Kuala Lumpur, Malaysia. Further studies are required to determine the possible source and effect of low-dose heavy metals exposure, as well as other environmental factors, on renal function.

### Funding

This study was supported by the National University of Malaysia (UKM) (UKM FPR.4/244/FF-2019-100, 22nd March 2019). The funders had no role in study design, data collection and analysis, decision to publish, or preparation of the manuscript.

### Grant Disclosures

The following grant information was disclosed by the authors:
The National University of Malaysia (UKM): UKM FPR.4/244/FF-2019-100, 22nd March 2019.

### Competing Interests

The authors declare there are no competing interests.

## Author Contributions

- Mohd Faizal Madrim conceived and designed the experiments, performed the experiments, analyzed the data, prepared figures and/or tables, authored or reviewed drafts of the paper, and approved the final draft.
- Mohd Hasni Ja'afar and Rozita Hod conceived and designed the experiments, authored or reviewed drafts of the paper, and approved the final draft.

## Human Ethics

The following information was supplied relating to ethical approvals (i.e., approving body and any reference numbers):

The National University of Malaysia (UKM) granted ethical approval to carry out the study (Ethical Approval Ref: UKM FPR.4/244/FF-2019-100, 22nd March 2019).

## Data Availability

Collected data are available as a Supplemental Files.

## Supplemental Information

Supplemental information for this article can be found online at http://dx.doi.org/10.7717/peerj.12014#supplemental-information.

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
