# Peer review of "Prevalence of abnormal urinary cadmium and risk of albuminuria as a primary bioindicator for kidney problems among a healthy population"

_PeerJ, doi:10.7717/peerj.12014_

## Round 0.1 · original submission · Major Revisions

The selection criteria for the study should be presented in more detail. Please, also, address the reviewers' comments.

Reviewers have directed your attention to or requested that you cite specific references. You may add them you believe they are especially relevant. However, I do not expect you to include these citations, and if you do not include them, this will not influence my decision.

Reviewer 1 ·

Basic reporting

no comment

Experimental design

no comment

Validity of the findings

Authors should measure the sensitivity of the analysis.

Additional comments

This study aimed to determine urine cadmium levels and their link to albuminuria as an early sign of renal disease in the Kepong community. In general, this plan is an interesting and well-written manuscript. Therefore, I recommend the publication after minor revisions as the following:
1- Please, update the prevalence of chronic kidney disease.
2- Please, refer to B40, M40, and T20 categories.
3- Is the range of age in this study similar to Adnan et. al., (2012)?
4- Could authors add sections for inclusions and exclusions criteria for this study.
5- Authors should measure the sensitivity of the analysis.
6- Authors should discuss is there any relationship between albuminuria and metabolic diseases in this study.
7- Please mention age as having a substantial influence on nephrotoxicity in the discussion section.
Reference: El-Arabey AA. Sex and age differences related to renal OCT2 gene expression in cisplatin-induced nephrotoxicity. Iran J Kidney Dis. 2015 Jul;9(4):335-6.

Reviewer 2 ·

Basic reporting

-

Experimental design

-

Validity of the findings

-

Additional comments

This is an interesting study. I have no significant objections. Some parts of the manuscript were repeated unnecessarily. Please consider deleting them, particularly in the Abstract. The part concerning cadmium must be much more strongly written (line 60, 61); I suggest the authors to carefully consider the papers of Stojsavljević et al., Jagodić et al.

---

## Round 0.2 · accepted · Accept

As all the comments from the reviewers were addressed to their satisfaction, the new version of the manuscript was significantly improved.

Reviewer 1 ·

Basic reporting

The authors have successfully addressed all issues

Experimental design

The authors have successfully addressed all issues

Validity of the findings

The authors have successfully addressed all issues

Additional comments

The authors have successfully addressed all issues

Reviewer 2 ·

Basic reporting

-

Experimental design

-

Validity of the findings

-

Additional comments

No remarks.